# Influence of Canopy Interception and Rainfall Kinetic Energy on Soil Erosion under Forests

**Guijing Li [1,2], Long Wan [1,2], Ming Cui [3], Bin Wu [1,2,*] and Jinxing Zhou [1,2]** 

[1]   Jianshui Research Station, School of Soil and Water Conservation, Beijing Forestry University, Beijing 100083, China; lgj8023lhy@163.com (G.L.); wanlong255@sina.com (L.W.); zjx001@bjfu.edu.cn (J.Z.)

[2]   Key Laboratory of State Forestry Administration on Soil and Water Conservation, Beijing Forestry University, Beijing 100083, China

[3]   Institute of Desertification Studies, Chinese Academy of Forestry, Beijing 100091, China; cuiming4057@126.com

*   Correspondence: wubin@bjfu.edu.cn; Tel.: +86-10-62336788

**Abstract:** Afforestation is a widely accepted measure to control soil erosion around the world. A large area of forest has been built to prevent slope soil erosion in the red soil region of southern China since the 1980s. The vegetation coverage has significantly increased; however, there is still moderate or severe soil erosion under the forest. In order to improve the situation, it is necessary to study the effects of canopy on soil erosion under the forest. Standard runoff plots were established on two typical sites, which represented pure *Pinus massoniana* Lamb. forest and bare land, respectively. Precipitation redistribution and throughfall indices including raindrop size, raindrop velocity, and the kinetic energy (KE) of raindrops were quantified. The results showed that 29.3% of the precipitation was directly prevented from reaching the forest land surface. The canopy interception effect was better under low rainfall intensity than high rainfall intensity. Compared with open rainfall, throughfall raindrops were 16.3% fewer in number, larger in size, and the range of throughfall drop size distribution (DSD) was enlarged. The volume ratio of large drops was larger with higher rainfall intensity. When the rainfall intensity was less than 14 mm h$^{-1}$, throughfall kinetic energy (TKE) was higher than open rainfall kinetic energy (OKE) owing to the higher volume ratio of large raindrops. When the rainfall intensity was more than 14 mm h$^{-1}$, TKE was smaller owing to the large raindrops failing to reach their final velocities: their mean velocity was 80% of their final velocity. The sediment yield was the largest under high rainfall intensity and the effect of sediment reduction was the largest under moderate rainfall intensity. Therefore, the largest KE did not lead to the maximum sediment yield; canopy interception was also an important factor affecting sediment yield.

**Keywords:** canopy interception; kinetic energy; *Pinus massoniana* Lamb.; soil loss under forest; sediment reduction rate

## 1. Introduction

Red soils, classified as Plinthosols in the World Reference Base for Soil Resources and characterized by acidity, nutrient deficiency, poor organic matter levels, and high erodibility, are widely studied in the world [1]. In China, red soil occupies approximately 2.04 million km$^2$ in tropical and subtropical regions [2,3]. The red soil region of southern China has faced severe soil erosion on account of high precipitation, mountainous and hilly landforms, and various irrational human activities [4,5]. It has been widely believed that vegetation, such as forests and grasslands, are key factors to control the erosion [6–9]. In order to effectively protect the surface soil from erosion caused by rainfall, an afforestation program with a large area has been put into practice since the 1980s, and masson pine

(*Pinus massoniana* Lamb.), as a good pioneer tree for forestation, now covers 30.5% of the total forest area of the region [10]. However, soil erosion under the forest in the area is not fully controlled, moderate or severe soil erosion is still observed [11–13]. To better control the soil erosion of red soil regions in China, it is urgent to solve the problem of forests being unable to reduce soil erosion effectively and to determine what the forest's mechanism of influence is on soil erosion after afforestation.

In forests, the canopy changed rainfall kinetic energy (KE) by intercepting precipitation, and thus, modifying raindrop size and velocity [14]. Throughfall kinetic energy (TKE) is a usually used indicator to express the potential of rainfall erosivity and predict soil erosion rates [15,16]. Brandt [17] first developed a model to calculate TKE and it was widely assumed that a forest canopy mitigated rainfall KE until Chapman [18] demonstrated forests could prominently increase TKE under some specific conditions. Numerous studies also confirmed that monoculture plantations increased TKE and accelerated soil erosion [15,19,20]. However, TKE is influenced by many factors including drop size distribution (DSD) [21–23], canopy architecture (e.g., canopy thickness, height) [24,25], tree species [14,22,25], even meteorological factors [14,22], etc., which indicate that TKE may vary greatly under different conditions.

As a zonal and good pioneer tree for forestation, masson pine is an irreplaceable coniferous afforestation species in the red soil region of southern China. Whether the masson pine forest is effective in reducing TKE and the relationship between TKE and accelerated soil erosion existed in the region after large-scale afforestation, but so far, this topic has rarely been studied. Such studies would provide a reference for management strategies for soil protection and rehabilitation and would further improve knowledge of mechanisms that determine soil erosion processes.

In this study, two representative types of standard runoff plots were built, one was bared as check plot and the other was planted with pure masson pine. The aims of the research were to (1) quantify canopy interception of masson pine forest; (2) analyze the influence of the canopy on rainfall KE; and (3) clarify the causes of moderate and severe soil erosion under a masson pine forest and to confirm its effect on soil and water conservation.

## 2. Material and Methods

### 2.1. Study Area

The study area was located at Changting County, Longyan City, Fujian Province, China, which has a geographic position ranging from 116°00′45″ to 116°39′20″ E, and from 25°18′40″ to 26°02′05″ N. It has a subtropical monsoon climate. The mean annual rainfall was 1700 mm during 1995–2014. The average annual temperature is 17.5 °C. Rainfall is mainly concentrated from April to August, during which rainfall accounts for more than 60% of the annual rainfall. The main soil type is red soil derived from granite and gneiss, the soil is as deep as 1 m. The study area belongs to the southern Wuyi Mountain; mountains and hills occupy 71% of the total area with steep slopes, and the valleys are mostly of "V" shape. The content of soil organic matter ranged from 0.95 to 8.51 g kg$^{-1}$. The soil texture was mainly sand, with a sand content of more than 50%; consequently, the soil is extremely vulnerable to erosion. Thus, the soil is barren owing to the long history of soil erosion.

Two sites were selected and the distance between them was 1 km. One served as a control with no vegetation at all, the other was planted with masson pine in 2004. Standard runoff plots were set up at the two sites (Figure 1). The length of the standard runoff plot was 20 m (horizontal projection), the width was 5 m (parallel to the contour line), the slope was 15°. The vertical projection area equaled 100 m$^2$. As for the masson pine runoff plots, there was little vegetation under the forest, and the stand density was 1500 plants per hectare, the average tree age was 10 years, the average tree height was 7.4 m, the average bottom height was 3.1 m, the canopy closure rate was 0.55, the average diameter at breast height (DBH) was 7.3 cm.

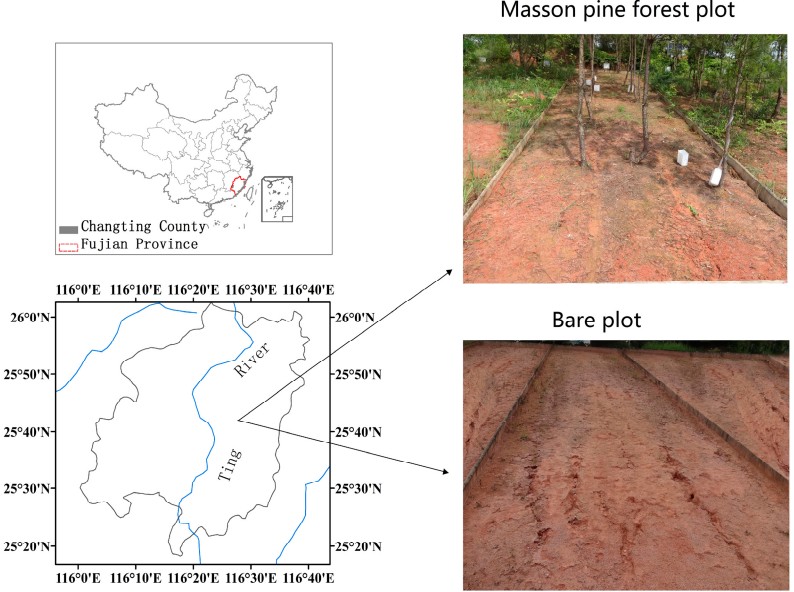

**Figure 1.** Study site locations and views of the studied plots.

## 2.2. Data Collection and Calculation

### 2.2.1. Open Rainfall and Canopy Interception

We measured open rainfall, throughfall, and stem flow for different rainfall events (Table 1) during the rainy season. Data from the Changting Hydrographic Bureau 10 year rainfall events showed that events with precipitation < 20 and 35 mm comprised 82.8% and 91.8% percent of the total rainfall events. Events with rainfall intensity < 20 and 45 mm comprised 92.9% and 97.4% percent of the total rainfall events. Therefore, the events we measured were typical. The volumetric moisture content of surface soil preceding the occurrence of rain ranged from 0.126 to 0.257 in the bare plot and from 0.159 to 0.305 in the forest plot.

**Table 1.** Rainfall events.

| Rainfall Events | Average Rainfall Intensity/mm h$^{-1}$ | Rainfall/mm | Rainfall Duration/min | Throughfall/mm | Stem Flow/mm | Canopy Interception/mm |
|---|---|---|---|---|---|---|
| 1 | 2.1 | 2.1 | 60 | 0.7 | 0.0 | 1.4 |
| 2 | 2.4 | 3.1 | 78 | 1.4 | 0.0 | 1.7 |
| 3 | 3.5 | 4.1 | 70 | 1.8 | 0.1 | 2.2 |
| 4 | 4.8 | 30.1 | 375 | 24.0 | 0.6 | 5.6 |
| 5 | 5.1 | 5.1 | 60 | 3.0 | 0.1 | 2.1 |
| 6 | 5.3 | 14.0 | 160 | 9.7 | 0.2 | 4.1 |
| 7 | 6.8 | 3.4 | 30 | 2.0 | 0.0 | 1.4 |
| 8 | 13.3 | 10.0 | 45 | 6.5 | 0.2 | 3.3 |
| 9 | 14.4 | 16.8 | 70 | 12.3 | 0.3 | 4.2 |
| 10 | 16.7 | 12.5 | 45 | 7.5 | 0.2 | 4.8 |
| 11 | 19.1 | 9.6 | 30 | 6.1 | 0.2 | 3.3 |
| 12 | 20.3 | 11.9 | 35 | 8.1 | 0.2 | 3.6 |
| 13 | 28.2 | 16.4 | 35 | 10.8 | 0.2 | 5.4 |
| 14 | 31.3 | 23.5 | 45 | 17.8 | 0.4 | 5.3 |
| 15 | 40.8 | 13.6 | 20 | 10.0 | 0.2 | 3.4 |

Precipitation was measured with Campbell TB4MM rain gauges, for which the sampling area was 0.031 m$^2$. Throughfall was collected from three buckets under the canopy of masson pine, which were distributed on the up, middle, and down slope of the plot. The top area of the bucket was 0.035 m$^2$ with a length of 0.25 m and a width of 0.14 m. The volume of collected rainwater measured with a graduated cylinder with a 1 mL accuracy divided by the top area of a bucket was equal to throughfall.

We took the mean as throughfall and its variable coefficient was 1.6%–15.9%. Three masson pines with similar DBH (Table 2) were selected in masson pine plots for stem flow sampling. Stem flow was collected from a bucket on the ground which was connected with a cut pipe made of polyethylene plastic (about 2 cm in diameter). The pipe was wrapped down along the position of the DBH of the trunk. The bottom area of the bucket was 0.035 m². The volume of collected rainwater measured by graduated cylinder with 1 mL accuracy divided by the bottom area of the bucket was equal to the stem flow. The variable coefficient of the stem flow was 2.0%–20.0%. Canopy interception was calculated by the water balance equation.

**Table 2.** Measurement index of the selected trees.

|   | Tree Age/a | Tree Height/m | Diameter at Breast Height/cm | Height at the Middle of Crown/m | Crown Width/m×m |
|---|---|---|---|---|---|
| 1 | 10 | 7.6 | 7.4 | 5.4 | 3.2 m × 3.5 m |
| 2 | 10 | 7.5 | 6.9 | 4.8 | 3.1 m × 3.1 m |
| 3 | 10 | 7.3 | 7.1 | 5.2 | 3.1 m × 3.2 m |

Notes: Because of the high variation in canopy thickness, we took the height at the middle of the crown, which was the distance from the canopy center to the ground, as the average falling height of raindrops under masson pine.

### 2.2.2. Raindrop Sampling and Measurement

Raindrop size was measured by the filter paper staining method [26]. In order to improve accuracy, the relationship between diameter of the dye stain and the raindrop size was re-calibrated, the details are as follows. First, the qualitative filter paper produced in Hangzhou, China with a diameter of 15 cm was evenly colored with pigment which was a mixture of eosin and talc powder in a mass ratio of 1:10. Second, medical needles with 0.45 mm, 0.6 mm, 0.7 mm, 1.2 mm, 1.6 mm in diameter were used to simulate the different diameters of raindrops. The total mass (M) of the syringe with water was weighed, then water was dropped over the filter paper, and the mass (m) of syringe with water was weighed again after the filter was filled with water drops. Third, the diameter of a water drop was:

$$\mathrm{dw} = \sqrt[3]{\frac{6(\mathrm{M} - \mathrm{m})}{n\rho\pi}}, \tag{1}$$

where dw is the diameter of a water drop, $n$ is the number of water drops falling on the filter paper, $\rho$ is the water density, 1 g cm$^{-3}$.

Finally, the diameter of the stain on the filter paper was measured by a Vernier caliper. The relationship between the diameter of water droplets and its corresponding stain size was obtained by regression analysis, which was:

$$\mathrm{d} = 0.42\,\mathrm{D}^{0.70}, R^2 = 0.967, \tag{2}$$

where d is the diameter of a water droplet (mm) and D is the size of the stain.

Three masson pines with similar DBH and crown width (Table 2) were selected in masson pine plots for raindrops sampling, and we selected 16 sampling points under each masson pine which were distributed in eight directions (Figure 2). There were two points in each direction evenly established at the canopy projection onto the ground [27]. As soon as the rainfall begun, raindrops sampling was conducted simultaneously inside and outside the forest with prepared filter paper. The sampling interval was 2 min. Raindrop sampling was conducted immediately if rainfall intensity suddenly changed. The number of stain papers we used in this study was 662 for the bare plot and 2528 for the forest plot.

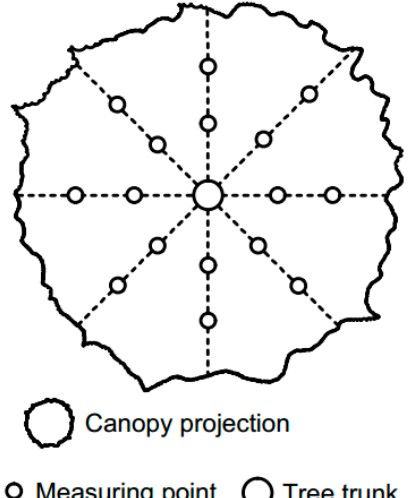

**Figure 2.** Measuring points under the canopy.

### 2.2.3. Raindrop Velocity

Raindrop terminal velocity (m s$^{-1}$) was calculated as [28]:

$$Vo = 0.0561\ d^3 - 0.912\ d^2 + 5.03\ d - 0.254, \tag{3}$$

Due to the shorter fall height, throughfall drops' velocities tend to be lower than Vo [29]. Throughfall drops' velocities (m s$^{-1}$) was calculated as [30]:

$$Vi = Vo\ (1 - \exp(-2gh/Vo^2))^{1/2}, \tag{4}$$

where $g$ is acceleration by gravity, 9.8 m s$^{-2}$, h is the falling height of throughfall drops (m), the average height in our experiment was 5.1 m.

### 2.2.4. Kinetic Energy of Drops and Soil Loss

The kinetic energy (J) of an individual raindrop was calculated as:

$$e = 1/2\ mv^2 = \rho\pi d^3 v^2/12, \tag{5}$$

where v is the raindrop velocity (m s$^{-1}$).

Unit kinetic energy (J m$^{-2}$ mm$^{-1}$) was calculated as:

$$Eu = \sum_{i=1}^{n} e_i/(P \times S), \tag{6}$$

where $n$ is the drop number, $P$ is precipitation that falls on the filter paper (mm), which equals the total mass of raindrops divided by raindrop density and filter paper area. $S$ is the area of a piece of filter paper (m$^2$). Eu could eliminate the effect of precipitation and rain rate.

We used a bucket for collecting runoff and sediment from study plots, which were stirred evenly when runoff was over, and the mixture of runoff and sediment was sampled by a bottle of certain volume with three repetitions. Filtering the collected samples, we then obtained the soil loss amount after drying and weighing the sediment.

## 3. Results

### 3.1. Influence of Canopy on Rainfall Re-Distribution

The total rainfall amount was 176.2 mm during the whole observation period; throughfall, stem flow, and canopy interception were 121.7 mm, 2.9 mm, and 51.6 mm (Table 1), accounting for 69.1%, 1.6%, and 29.3% of the open rainfall. It meant that 70.7% of the open rainfall reached the soil surface under the forest.

Amounts of throughfall were greater with higher rainfall intensity, additionally, the growth rates were almost stable which can be fitted by a power function (Figure 3a). Amounts of stem flow and canopy interception first increased with increasing rainfall intensity, meanwhile, the growth rates gradually decreased with the increased levels of rainfall intensity and then tended to be constant which can be fitted by a log function (Figure 3b,c). Percentage of throughfall and stemflow were erratic below 7 mm h$^{-1}$ and then more or less a plateau (Figure 3a,b). The percentage of canopy interception first rapidly decreased with increasing rainfall intensity and then became relatively constant (Figure 3c), which implied that the percentage of canopy interception was the largest for events of low rainfall intensity (<7 mm h$^{-1}$). In turn, the percentage of rainfall reaching the soil surface under the forest was the largest for events of high rainfall intensity (>30 mm h$^{-1}$). Moreover, rainfall events with high rainfall intensity tended to have short rainfall duration (Table 1). Therefore, rainfall events with high intensity and short duration may produce more precipitation reaching the forest land surface than rainfall events with low intensity and long duration under the condition of the same rainfall amount.

Based on the fitted functions, stem flow generated when rainfall intensity was higher than approximately 2.1 mm h$^{-1}$. The constant value of the stem flow and canopy interception amounts were 0.26 mm and 4.6 mm, and the constant value of the percentages were 1.9% and 27.1%, respectively.

(a)

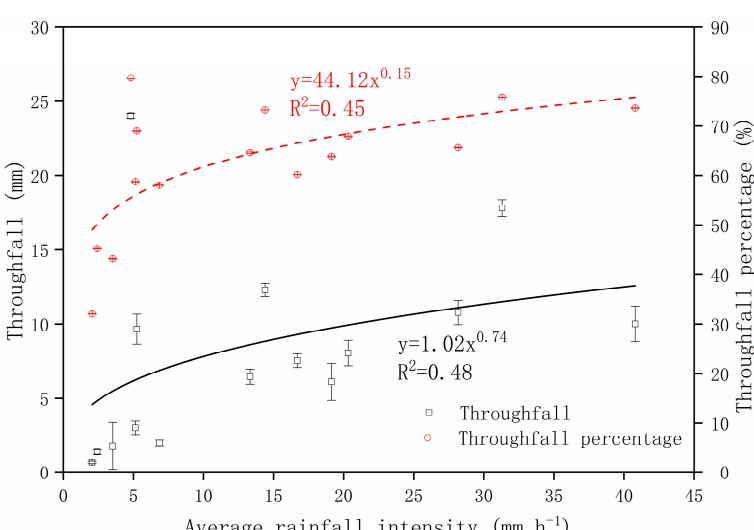

**Figure 3.** *Cont.*

(b)

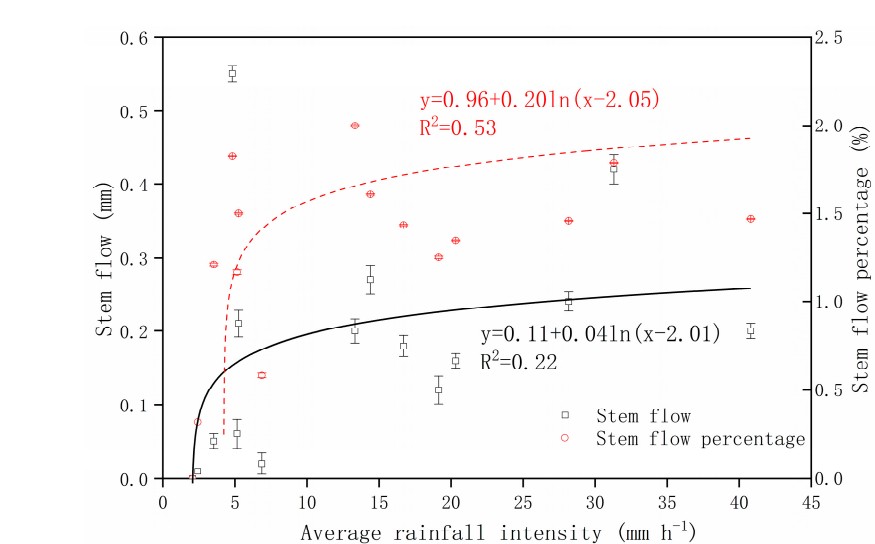

(c)

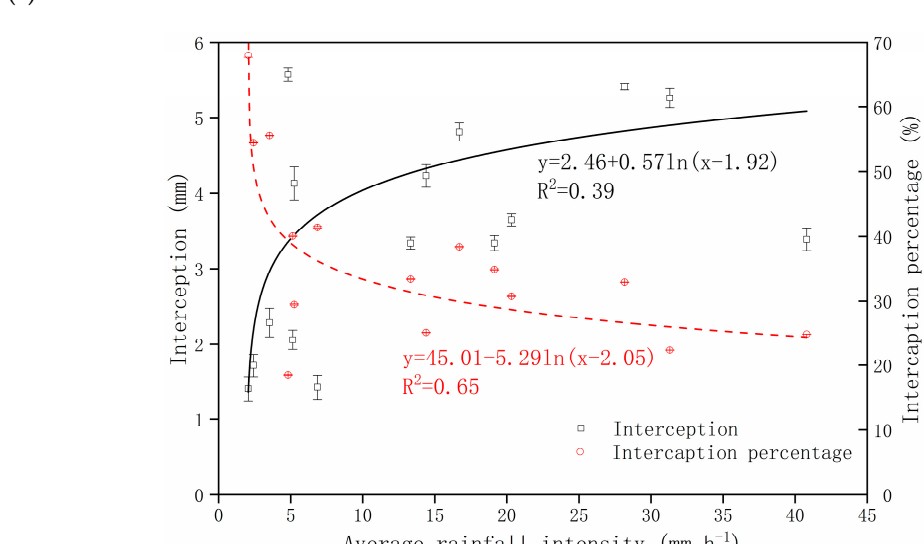

**Figure 3.** Rainfall redistribution as a function of rainfall intensity. (**a**) Distribution of throughfall and its percentage with rainfall intensity; (**b**) Distribution of stem flow and its percentage with rainfall intensity; (**c**) Distribution of interception and its percentage with rainfall intensity.

*3.2. Influence of the Canopy on Characteristics of Raindrops*

Raindrops inside and outside the forest under five different rainfall events were measured, including one rainfall event with low rainfall intensity, three rainfall events with moderate rainfall intensity (7~30 mm h$^{-1}$), and one rainfall event with high rainfall intensity (Table 3).

**Table 3.** Drop size distribution (DSD) information for different rainfall events.

| Rainfall Intensity/ mm h$^{-1}$ | Drop Number | | DSD Range | | D$_{50}$/mm | |
|---|---|---|---|---|---|---|
| | Open | Throughfall | Open/mm | Throughfall/mm | Open/mm | Throughfall/mm |
| 5.1 | 2990 | 2560 | 0.59~2.60 | 0.35~2.89 | 1.28 | 1.64 |
| 13.3 | 2681 | 2264 | 0.66~2.75 | 0.56~3.14 | 1.47 | 1.75 |
| 19.1 | 1986 | 1697 | 0.70~3.63 | 0.58~4.62 | 1.88 | 2.20 |
| 28.2 | 2116 | 1792 | 0.76~5.01 | 0.62~5.49 | 2.60 | 3.12 |
| 40.8 | 1498 | 1127 | 0.95~5.71 | 0.63~6.08 | 2.94 | 3.90 |

The total number of raindrops observed for open rainfall was 11,271, whereas it was 9440 for throughfall, 16.3% fewer than that of open rainfall. The median volume of the drop diameters ($D_{50}$) for open rainfall raindrops was 2.00 mm, and the $D_{50}$ of throughfall raindrops was 3.18 mm (Figure 4a), 1.59 times as much as that of open rainfall. The $D_{50}$ of open rainfall and throughfall were larger with higher rainfall intensity (Table 3, Figure 4a). The raindrop diameter of open rainfall ranged from 0.59 to 5.71mm, and the raindrop diameter of throughfall ranged from 0.35 to 6.08 mm. Compared with open rainfall, the throughfall raindrops were larger in size, and the range of throughfall–DSD was enlarged.

There was a clear DSD difference between open rainfall and throughfall (Figure 4b). The DSD of open rainfall was unimodal, whereas throughfall–DSD was bimodal, and the peaks corresponding to drop diameters were greater with higher rainfall intensity; however, the peaks corresponding to volume ratio were smaller with higher rainfall intensity. The peak of open rainfall corresponding to drop diameter increased from 1.17 to 2.60 mm, the peak corresponding to volume ratio decreased from 12.4% to 7.6%. The first peak of throughfall corresponding to drop diameter increased from 0.76 to 2.04 mm, and the peak corresponding to volume ratio decrease from 5.4% to 4.3%. The second peak of throughfall corresponding to drop diameter increased from 1.88 to 4.08 mm, and the peak corresponding to volume ratio decreased from 8.2% to 6.5%. Small droplets (<1 mm) accounted for 44.90% in number and 3.05% in volume. Large drips (>3 mm) accounted for 6.09% in number but 57.72% in volume. Our findings indicated that raindrop volume was more influenced by raindrop size than by raindrop number.

Figure 5 indicated the influence of rainfall intensity on DSD. The volume ratios of small and medium droplets (1–3 mm) were smaller with higher rainfall intensity. Conversely, the volume ratio of large drops was larger with higher rainfall intensity. The mean volume ratio of large drops for open rainfall was 0.26 (range from 0 to 0.45), whereas throughfall was 0.58 (range from 0 to 0.64), which was greater than that from open rainfall. The results were similar to the conclusion of Tsukamoto [31]. Figure 5 also showed that there were no large raindrops for open rainfall until rainfall intensity reached 19.1 mm $h^{-1}$, whereas the value for throughfall was 13.3 mm $h^{-1}$. After rainfall passed through the canopy, the threshold of rainfall intensity for large raindrops generation decreased.

(a)

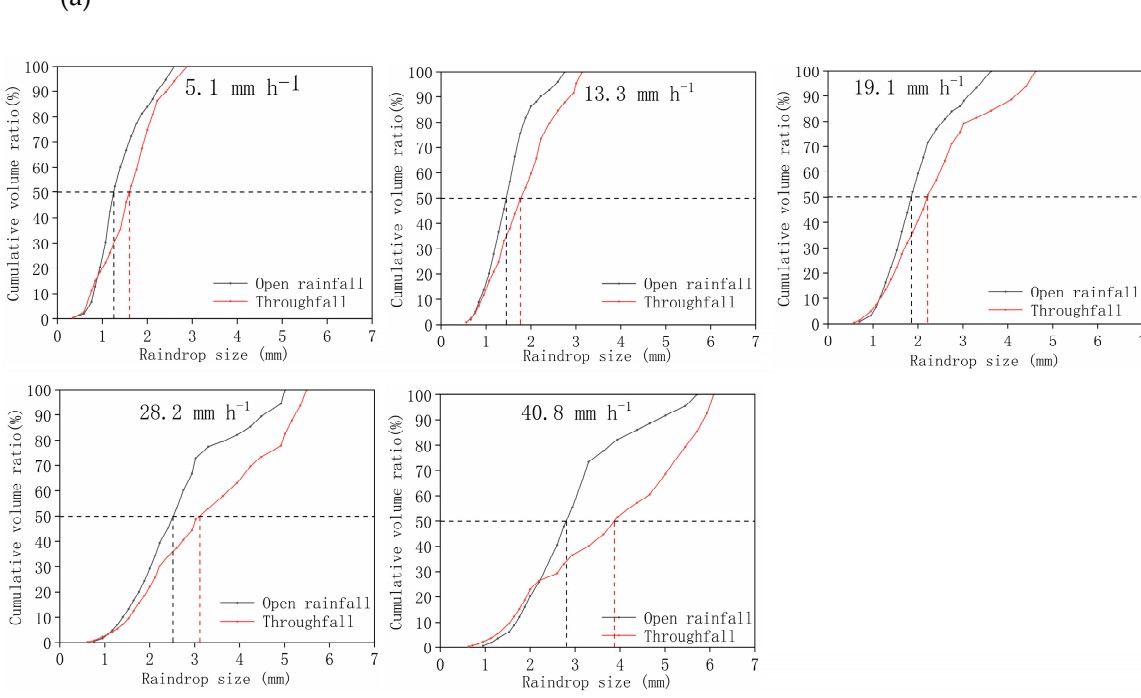

**Figure 4.** *Cont.*

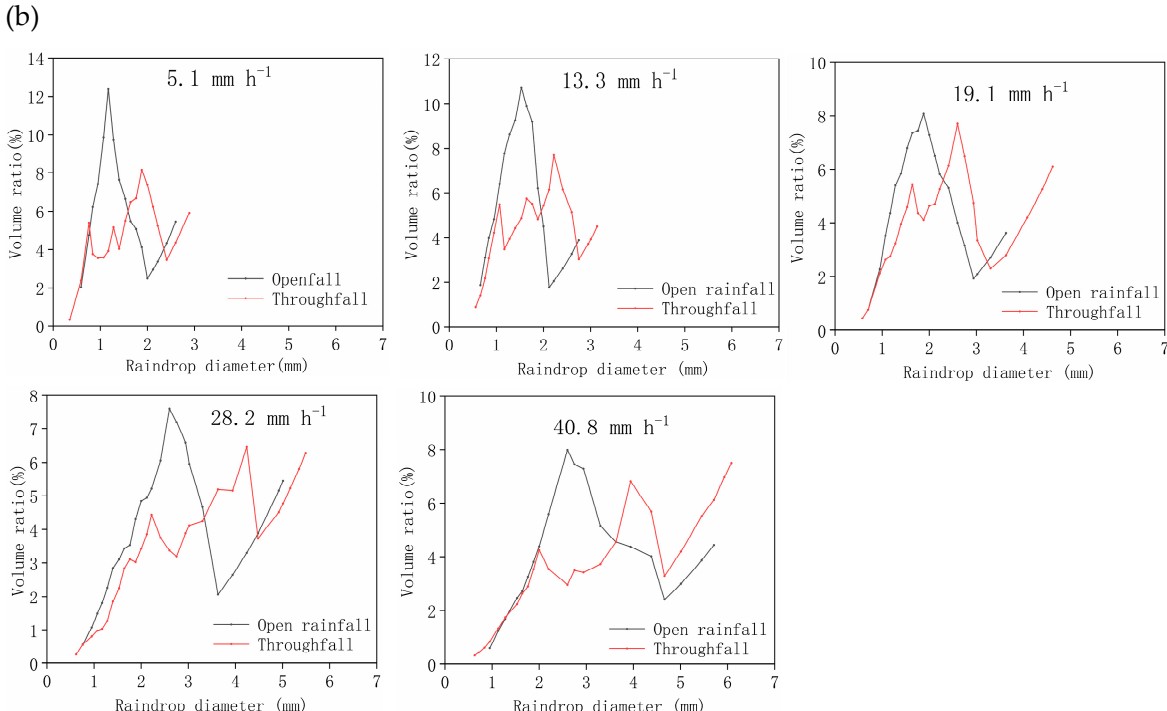

**Figure 4.** (**a**) Cumulative DSD for open rainfall and throughfall. The dashed, horizontal lines indicate the 50% lines of cumulative DSD. The dashed, vertical lines represent the $D_{50}$. (**b**) DSD for open rainfall and throughfall.

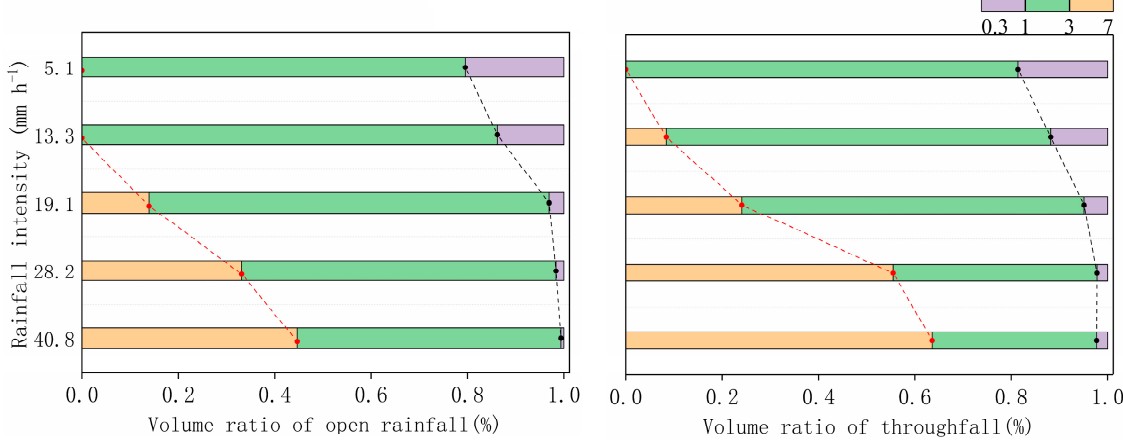

**Figure 5.** Volume ratio of raindrops with different size raindrops under different rainfall intensity. The dashed, red lines indicate the volume ratio of large drops, and the dashed, black lines represent the volume ratio of small drops.

### 3.3. Influence of Canopy on Rainfall Kinetic Energy

The calculated open rainfall kinetic energy (OKE) under different rainfall intensity ranged from 13.91 to 30.69 J m$^{-2}$ mm$^{-1}$, while TKE ranged from 15.03 to 25.08 J m$^{-2}$ mm$^{-1}$. Both of them were in the normal range (ranged from 11 to 36 J m$^{-2}$ mm$^{-1}$) derived from the analysis of measurements from around the world [26]. The mean OKE and TKE were 24.09 and 23.68 J m$^{-2}$ mm$^{-1}$ respectively, and the latter was 1.71% smaller than the former. With respect to the effect of rainfall intensity on KE, the relationship between them was an exponential function (Figure 6), and the regression formulas were as follows:

$$OKE = 28.74\,(1 - \exp(-0.11x))\ (R^2 = 0.70), \tag{7}$$

$$\text{TKE} = 23.07 \ (1 - \exp(-0.14x)) \ (R^2 = 0.61), \tag{8}$$

There was a threshold indicating the magnitude of KE between open rainfall and throughfall (Figure 6). When the rainfall intensity was less than 14 mm h$^{-1}$, TKE was higher than OKE, which means high risk of erosion under forest. The effect was reversed when the rainfall intensity was more than 14 mm h$^{-1}$.

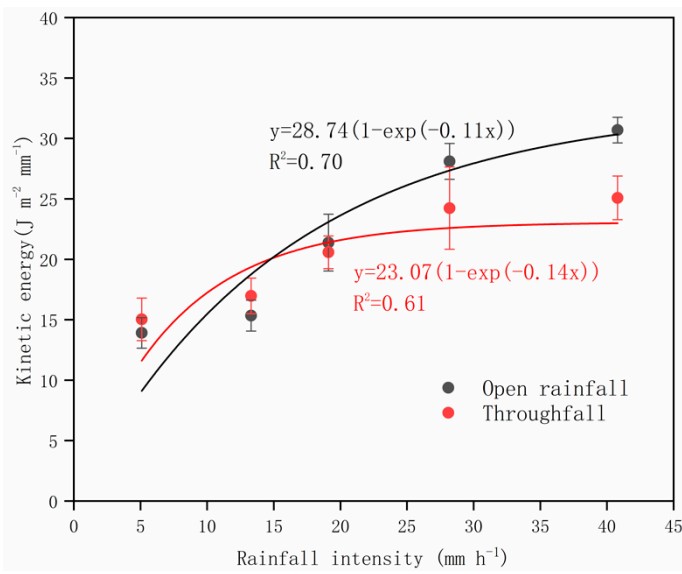

**Figure 6.** Scatter plot of kinetic energy versus rainfall intensity with exponential law.

### 3.4. Comparison of Soil Loss Inside and Outside the Forest

Table 4 showed that soil loss of bare plots and forest plots were both higher with greater total KE of raindrops except for the rainfall event with high rainfall intensity. We made a correlation analysis between KE of three different classes of raindrops and sediment yield (Table 5), the results showed that the correlation between KE of large raindrops and sediment yield was stronger than the other two classes of raindrops, and the correlation coefficient between the KE of large raindrops and sediment yield in the bare plot was 0.94. The correlation coefficient between the KE of large raindrops and sediment yield in the forest plot was 0.80. Soil loss in the forest plot was significantly smaller than that in the bare plot and the average sediment reduction rate was 62.6% (ranging from 50.3% to 71.8%). Meanwhile, the effect of the forest on soil and water conservation was different under different rainfall intensities. The dynamic characteristics were that the effect of sediment reduction first increased and then decreased with the increase of rainfall intensity. The effect of sediment reduction was similar under low and high rainfall intensities, and the difference between them was 5.3%. The effect of sediment reduction was the largest under moderate rainfall intensity.

**Table 4.** Soil loss of two different types of plots under different rainfall events.

| Rainfall Intensity/mm h$^{-1}$ | Bare Plot | | | | | Masson Pine Forest Plot | | | | |
|---|---|---|---|---|---|---|---|---|---|---|
| | Total KE/J m$^{-2}$ | | | | Soil Loss/g | Total KE/J m$^{-2}$ | | | | Soil Loss/g |
| | <1 mm | 1–3 mm | >3 mm | Total | | <1 mm | 1–3 mm | >3 mm | Total | |
| 5.1 | 5.68 | 65.26 | 0.00 | 70.94 | 95.2 ± 1.10 | 2.71 | 42.38 | 0.00 | 45.09 | 47.3 ± 1.04 |
| 13.3 | 7.78 | 145.62 | 0.00 | 153.40 | 403.8 ± 1.61 | 4.21 | 91.80 | 14.30 | 110.31 | 149.7 ± 3.12 |
| 19.1 | 1.84 | 155.61 | 47.80 | 205.25 | 2027.4 ± 2.12 | 1.70 | 82.66 | 41.18 | 125.54 | 620.1 ± 8.10 |
| 28.2 | 1.68 | 250.10 | 209.06 | 460.84 | 3967.9 ± 1.82 | 1.33 | 89.71 | 170.64 | 261.68 | 1118.0 ± 10.79 |
| 40.8 | 0.59 | 186.09 | 230.71 | 417.38 | 5825.9 ± 17.93 | 1.26 | 65.93 | 183.61 | 250.80 | 2678.0 ± 6.68 |

**Table 5.** The correlation between the KE of different classes of raindrops and soil loss.

| Total KE/J m$^{-2}$ | Soil Loss in Bare Plot/g | Soil Loss in Forest Plot/g |
| --- | --- | --- |
| Small raindrops (<1 mm) | y = 5950.5 exp (−0.474x), $R^2$ = 0.72 | y = 3672.3 exp (−0.967x), $R^2$ = 0.56 |
| Medium raindrops (1–3 mm) | y = 31.0 exp (0.022x), $R^2$ = 0.76 | y = 38.7 exp (0.032x), $R^2$ = 0.17 |
| Large raindrops (>3 mm) | y = 671.2 exp (0.009x), $R^2$ = 0.94 | y = 110.5 exp (0.016x), $R^2$ = 0.80 |

## 4. Discussion

### 4.1. Influence of masson pine's Canopy on TKE

Previous studies have shown that KE would be higher with greater rainfall intensity [28,32,33]. Our results indicated that there was an exponential function relationship between KE and rainfall intensity, and the correlation coefficient was relatively high. OKE was higher under low rainfall intensity and TKE was higher under high rainfall intensity, owing to KE being not only related to DSD, but also to fall velocities of raindrops when it reached the ground (Figure 5, Figure 7). It was also suggested that the masson pine canopy had a more obvious mitigation effect on KE under high rainfall intensity in comparison with low rainfall intensity in the study area.

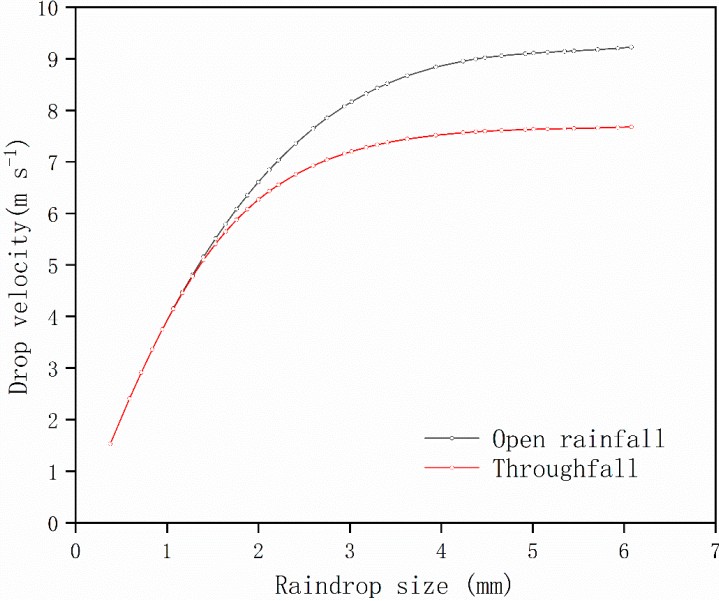

**Figure 7.** The relationship of raindrop velocity to diameter of total observed drops.

With respect to throughfall–DSD, it was generated from a combination of free throughfall–DSD and drip DSD [34]. It differed among tree species due to the variable plant surface characteristics [35]. $D_{50}$ is a widely used index representing both open rainfall and throughfall [28,36]. The $D_{50}$ under masson pine was 3.18 mm, smaller than that under sawtooth oak (*Quercus acutissima*), which was 3.60 mm [14]. In general, broad-leaved trees have larger $D_{50}$ than coniferous trees because rainwater could not coalesce readily on the fine leaves [27,37].

In our experiment, the free throughfall is falling with terminal velocity; however, the height of masson pine was insufficient for release throughfall gain final velocity. Considering that free throughfall accounted for only 30% of throughfall volume [35] and the KE of a large drop was much greater than the sum of the energies of a number of smaller drops [38], we took the height at the middle of crown (5.1 m) as the average falling height of throughfall raindrops. It may underestimated falling velocity of throughfall, thus, laser drop-sizing gauges should be used to confirm the influence of canopy height, thickness, and species on TKE in future studies.

Large drops were not observed under low rainfall intensity; however, the volume ratio of medium raindrops for throughfall was larger than that for open rainfall. Meanwhile, the difference was small between fall velocity and terminal velocity reaching the ground for medium raindrops (Figure 7). Their mean velocity was up to 95% of their final velocity, almost all the raindrops reached their terminal velocity. Therefore, TKE was greater owing to the higher volume ratio of large raindrops in comparison with OKE. However, although the volume ratio of large drops for throughfall was 42.4% higher than that for open rainfall under high rainfall intensity, the difference between fall velocity and terminal velocity was relatively large. Large raindrops generally reach their final velocity at a height of greater than 12 m [29]. The average falling height of raindrops in our experiment was 5.1 m, which was far from insufficient for large raindrops to gain final velocity. The mean velocity of large raindrops was 80% of their final velocity. Therefore, TKE was smaller owing to large raindrops failing to reach their final velocities in comparison with OKE.

### 4.2. Effect of masson pine Forest on Sediment Reduction

Soil erosion is caused by the detachment and transport of soil by raindrops and surface flow [6]. The KE of raindrops have an important effect on the initial process of soil erosion by splashing soil particles into the air [39]. Therefore, reducing the KE of raindrops was an important aspect in order to reduce soil erosion. In our experiment, throughfall was 69.1% of open rainfall on average (Table 1), the precipitation reaching the forest surface reduced. The total KE of throughfall was 39.3% smaller than that of open rainfall (Table 4). Thus, the sediment yield in the forest plot was less than that in the bare plot. In addition, we found that the sediment yield was not greater with higher total KE of rainfall, but total KE of large raindrops (Table 4). We suspected that impacts of large raindrops more easily caused soil splash detachment on the forest floor, and decreased infiltration capacity [40–42], which was similar to the view that the sediment yield in a rainfall event with high rainfall intensity and short duration was far greater than that in a rainfall event with low rainfall intensity and long duration [43–45]. But it did not have the best effect of soil and water conservation (Table 4) under the circumstances, the sediment reduction effect was the largest under moderate rainfall intensity.

Under the condition of low rainfall intensity, in spite of the canopy interception rate was the highest (Figure 3c), large raindrops were not observed inside and outside of the forest and the sediment yield inside and outside of the forest were relatively smaller, which were not on the same order of magnitude compared to that of the other two rainfall intensities. Under the condition of high rainfall intensity, 39.9% of total KE was buffered and 25.0% of the precipitation was intercepted by the canopy. Under the condition of moderate rainfall intensity, 39.3% of total KE was buffered and 33.3% of precipitation was intercepted by the canopy. The buffering effect of canopy was similar between moderate rainfall intensity and high rainfall intensity, but the canopy interception rate was 24.9% smaller under the condition of high rainfall intensity on account of the relatively large impact energy of the raindrops on the canopy, which gave rise to up to 75% of the precipitation directly reaching the land surface in the forest, and correspondingly, the effect of sediment reduction was relatively weaker.

However, there was still the problem that serious soil erosion remained with the erosion modulus of a rainfall event as high as 80.340 t km$^{-2}$ h$^{-1}$. Considering the fact that precipitation was abundant and were mainly storms with high rainfall erosivity in the red soil region of southern China, the erosion modulus may be greater than 2500 t km$^{-2}$ year$^{-1}$, the average erosion modulus of moderate and severe soil erosion put forward by the Ministry of Water Resources. It was suggested that soil erosion was generally severe if the surface coverage fell below 50% [46,47]. The ground cover reduced soil detachment by mitigating raindrop impact, and it could be more effective than tree canopy on reducing soil erosion in forest landscapes [15,48]. However, owing to large-scale aerial seeding in the 1980s in the study area, weak penetration of sunlight into the masson pine forest hindered the growth of understory vegetation and gave rise to bare ground, which exposed the soil surface to the direct impact of throughfall drops. Consequently, integrated forestry resource management measures were necessary to taken into account in the process of restoring degraded ecosystems including stand density, canopy

height, and structure, and the planting of understory vegetation to effectively eliminate soil erosion under forest.

Although observed rainfall events were limited, this study provides an important basis for a further understanding of soil erosion under the forest in the red soil region of southern China. It is also helpful for the design and management of soil erosion control measures in forests.

## 5. Conclusions

The sediment yield was largely dependent on rainfall intensity and KE of large raindrops in the area. $D_{50}$ and the DSD range of throughfall under masson pine was enlarged in comparison with that of open rainfall. TKE was higher than OKE when the rainfall intensity was less than 14 mm h$^{-1}$. The effect was reversed when the rainfall intensity was more than 14 mm h$^{-1}$. The rule of soil loss under the forest was revealed in the paper. In future research, the combination of rainfall erosivity and soil erosion is quite necessary. We also found that pruning and understory vegetation were important factors to prevent soil erosion during the forest management process.

**Author Contributions:** Conceptualization, G.L., B.W., M.C. and J.Z.; Data curation, G.L., M.C. and J.Z.; Formal analysis, G.L., M.C. and J.Z.; Funding acquisition, L.W. and J.Z.; Investigation, G.L. and L.W.; Supervision, B.W., J.Z., M.C. and L.W.; Validation, B.W., J.Z., M.C. and L.W.; Writing—original draft, G.L.; Writing—review and editing, G.L.

**Funding:** This research was funded by the National Key R&D Program of China (2016YFC0502504), the National Natural Science Foundation of China (31870707; 31700640) and the First-class Discipline Construction Project of Beijing Forestry University (2019XKJS0307). The APC was funded by 2016YFC0502504.

**Acknowledgments:** We would like to thank Jiufu Luo, Shuai Yan and Wei Zhou for their help with measurements in the field and laboratory. We also would like to thank the reviewers for comments on our paper and thank the editor for your advice.

**Conflicts of Interest:** The authors declare no conflicts of interest.

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
