# Peer review of "Influence of Canopy Interception and Rainfall Kinetic Energy on Soil Erosion under Forests"

_forests, doi:10.3390/f10060509_

Round 1
Reviewer 1 Report
General comments:
The study estimated rainfall partitioning by masson pines, throughfall kinetic energy, and sediment yield based on the comparison with open space. I am impressed you measured DSD by filter paper methods. Good work. Direct measurement of throughfall DSD is uncommon and their data is valuable. Analysis on the importance of large drops on sediment yield is interesting. However, the paper is immature for publication due to insufficient reviews of previous important studies (Introduction and Discussion) and insufficient information of methodologies (both measurement and calculation). Furthermore, the representativeness of measured rainfall events must be confirmed at first. The paper should be revised with them and major revision is necessary.
Major comments:
1.The paper missed recent review paper on throughfall DSD. Read it in detail at first. For example, you can compare D50 of masson pine with other species.
- Levia, D. F., Hudson, S. A., Llorens, P., & Nanko, K. (2017). Throughfall drop size distributions: a review and prospectus for future research. Wiley Interdisciplinary Reviews: Water, 4, e1225. https://doi.org/10.1002/wat2.1225
2.Most important paper on throughfall kinetic energy was missed. Studies by Brandt firstly developed model to calculate throughfall kinetic energy.
- Brandt, J. (1988). The transformation of rainfall energy by a tropical rain forest canopy in relation to soil erosion. Journal of Biogeography, 15, 41–48. Retrieved from http://www.jstor.org/stable/10.2307/2845044
- Brandt, C. J. (1989). The size distribution of throughfall drops under vegetation canopies. Catena, 16, 507–524. Retrieved from http://www.sciencedirect.com/science/article/pii/0341816289900325
- Brandt, C. J. (1990). Simulation of the size distribution and erosivity of raindrops and throughfall drops. Earth Surface Processes and Landforms, 15, 687–698. Retrieved from http://onlinelibrary.wiley.com/doi/10.1002/esp.3290150803/abstract
3.Reviewing is insufficient. Read and cite recent papers on throughfall DSDs, kinetic energy and plot study to measure soil erosion in forest, for example:
- Miyata, S., Kosugi, K., Gomi, T., & Mizuyama, T. (2009). Effects of forest floor coverage on overland flow and soil erosion on hillslopes in Japanese cypress plantation forests. Water Resources Research, 45(6). https://doi.org/10.1029/2008WR007270
- Goebes, P., Schmidt, K., Härdtle, W., Seitz, S., Stumpf, F., Oheimb, G. von, & Scholten, T. (2016). Rule-based analysis of throughfall kinetic energy to evaluate biotic and abiotic factor thresholds to mitigate erosive power. Progress in Physical Geography, 40(3), 431–449. https://doi.org/10.1177/0309133315624642
- Shinohara, Y., Ichinose, K., Morimoto, M., Kubota, T., & Nanko, K. (2018). Factors influencing the erosivity indices of raindrops in Japanese cypress plantations. CATENA, 171, 54–61. https://doi.org/10.1016/j.catena.2018.06.030
- Song, Z., Seitz, S., Zhu, P., Goebes, P., Shi, X., Xu, S., … Scholten, T. (2018). Spatial distribution of LAI and its relationship with throughfall kinetic energy of common tree species in a Chinese subtropical forest plantation. Forest Ecology and Management, 425(March), 189–195. https://doi.org/10.1016/j.foreco.2018.05.046
- Levia, D. F., Nanko, K., Amasaki, H., Giambelluca, T. W., Hotta, N., Iida, S., … Yamada, K. (2019). Throughfall partitioning by trees. Hydrological Processes. https://doi.org/10.1002/hyp.13432
4.L62. There are some studies to estimate throughfall KE under low canopy height. Add references here.
5.L88-95. Show tree height, canopy bottom height, stand density (or the number of trees trees in the plot), and canopy openness. They are useful for readers.
6.Section 2.2.1. Clarify your measurement methodology. Sampling area of the rain gauge, sampling area of the buckets for throughfall measurement, and the number of trees to measure stemflow. Also, did you measure sediment yield without stemflow input in the plot? If so, you should describe it.
7.Section 2.2.1. Clarify your calculation methodology. Did you use mean of three throughfall buckets? If so, how much was the variability? How did you convert stemflow volume [L] into [mm]?
8.Section 2.2.2. You selected totally 48 points (16 points x 3 trees) for the filter paper method. How many stain papers did you use in this study? The detail is not shown.
9.The total rainfall amount was 176.2 mm for your measurement, however is was SMALLER than annual rainfall, 1700 mm. Did the observed rainfall events (Table 2) have average characteristics? Do they have representativeness for this area? I cannot judge from the paper because characteristics of rainfall events in this area, such as ranges of average rainfall intensity, rainfall amount and rainfall duration, are not shown. For example, if you collected rainfall events with relatively smaller intensity or amount, your data will overestimate interception loss.
10.L258-262. This analysis is interesting. I recommend to add figure or table to show the importance of large drops on sediment yield.
11.Section 4.1. Your calculation is based that all of throughfall was falling from 5.1 m high. It underestimates falling velocity of throughfall because (1) free throughfall is falling with terminal velocity, (2) release throughfall is not always fallen from crown bottom but higher part of crowns (see references). The discussion is insufficient.
- Brandt, C. J. (1990). Simulation of the size distribution and erosivity of raindrops and throughfall drops. Earth Surface Processes and Landforms, 15, 687–698. Retrieved from http://onlinelibrary.wiley.com/doi/10.1002/esp.3290150803/abstract
- Nanko, K., Onda, Y., Ito, A., & Moriwaki, H. (2008). Effect of canopy thickness and canopy saturation on the amount and kinetic energy of throughfall: An experimental approach. Geophysical Research Letters, 35, L05401. https://doi.org/10.1029/2007GL033010
- Wakiyama, Y., Onda, Y., Nanko, K., Mizugaki, S., Kim, Y., Kitahara, H., & Ono, H. (2010). Estimation of temporal variation in splash detachment in two Japanese cypress plantations of contrasting age. Earth Surface Processes and Landforms, 35(9), 993–1005. https://doi.org/10.1002/esp
- Nanko, K., Onda, Y., Ito, A., & Moriwaki, H. (2011). Spatial variability of throughfall under a single tree: Experimental study of rainfall amount, raindrops, and kinetic energy. Agricultural and Forest Meteorology, 151, 1173–1182. https://doi.org/10.1016/j.agrformet.2011.04.006
- Song, Z., Seitz, S., Zhu, P., Goebes, P., Shi, X., Xu, S., … Scholten, T. (2018). Spatial distribution of LAI and its relationship with throughfall kinetic energy of common tree species in a Chinese subtropical forest plantation. Forest Ecology and Management, 425(March), 189–195. https://doi.org/10.1016/j.foreco.2018.05.046
Detailed comments:
L108. Bottom area is not correct. Top area is correct.
L118. What sizes did you use? Show them concretely.
L122. What is “dw”? No explanation.
L132. What was measured at “two sampling points”?
Table 2. Is “Height at the middle of crown” is the height between soil surface and crown bottom?
Figure 3. “a, b, c” are not shown in the figure. Caption of x-axis is “event average rainfall intensity”. Add error bar on each plot.
Figure 4. Use same range for x-axis, such as from 0 to 7 mm.
Figure 6. Power function is not appropriate unit kinetic energy reach plateau (van Dijk et al. 2002). Exponential is better.
Author Response
We would like to express sincere thanks for your affirmation of our work including uncommon measurement of throughfall DSD, valuable data and interesting analysis on the importance of large drops on sediment yield. Especially thank you for the detailed revision suggestions about the shortcomings of the paper and improvement for public. The following are the responses to the comments point to point.
Point 1: The paper missed recent review paper on throughfall DSD. Read it in detail at first. For example, you can compare D50 of masson pine with other species.
- Levia, D. F., Hudson, S. A., Llorens, P., & Nanko, K. (2017). Throughfall drop size distributions: a review and prospectus for future research. Wiley Interdisciplinary Reviews: Water, 4, e1225. https://doi.org/10.1002/wat2.1225
Response 1: I have read the reference you listed above carefully, and I have corrected in the Introduction: However, TKE was influenced by many factors including drop size distribution (DSD) [21-23], shown in L59 in manuscript with ‘Track Changes’ mode. I compared D50 of masson pine with other species in Section 4.1: The D50 under masson pine was 3.18 mm, smaller than that under sawtooth oak (Quercus acutissima), which was 3.60 mm [14]. In generally, broad-leaved trees have larger D50 than coniferous trees because rainwater could not coalesce readily on the fine leaves [27, 37]. The following line numbers were all in the manuscript with ‘Track Changes’ mode.
14. Nanko, K.; Hotta, N.; Suzuki, M. Evaluating the influence of canopy species and meteorological factors on throughfall drop size distribution. Journal of Hydrology. 2006, 329, 422-431.
21. Calder, I.R.; Hall, R.L.; Prasannab, K.T. Hydrological impact of Eucalyptus plantation in India. Journal of Hydrology. 1993, 150, 635-648.
22. Levia, D. F.; Hudson, S. A.; Llorens, P.; Nanko, K. Throughfall drop size distributions: a review and prospectus for future research. Wiley Interdisciplinary Reviews: Water. 2017, 4, e1225.
23. Zabret, K.; Rakovec, Jože; Mikoš, Matjaž; Šraj, Mojca. Influence of raindrop size distribution on throughfall dynamics under Pine and Birch trees at the rainfall event level. Atmosphere. 2017, 8, 240.
27. Nanko, K.; Onda, Y.; Ito, A.; Moriwaki, H. Spatial variability of throughfall under a single tree: Experimental study of rainfall amount, raindrops, and kinetic energy. Agricultural and Forest Meteorology. 2011, 151, 1173-1182.
37. Song, Z.; Seitz, S.; Zhu, P.; Goebes, P.; Shi, X.Z.; Xu, S.X.; Wang, M.Y.; Schmidt, K.; Scholten, T. Spatial distribution of LAI and its relationship with throughfall kinetic energy of common tree species in a Chinese subtropical forest plantation. Forest Ecology and Management, 2018, 425, 189-195.
Point 2: Most important paper on throughfall kinetic energy was missed. Studies by Brandt firstly developed model to calculate throughfall kinetic energy.
- Brandt, J. (1988). The transformation of rainfall energy by a tropical rain forest canopy in relation to soil erosion. Journal of Biogeography, 15, 41–48. Retrieved from http://www.jstor.org/stable/10.2307/2845044
- Brandt, C. J. (1989). The size distribution of throughfall drops under vegetation canopies. Catena, 16, 507–524. Retrieved from http://www.sciencedirect.com/science/article/pii/0341816289900325
- Brandt, C. J. (1990). Simulation of the size distribution and erosivity of raindrops and throughfall drops. Earth Surface Processes and Landforms, 15, 687–698. Retrieved from http://onlinelibrary.wiley.com/doi/10.1002/esp.3290150803/abstract
Response 2: I have corrected in L55 ‘Brandt [17] firstly developed a model to calculate TKE and it was widely assumed that a forest canopy mitigated rainfall KE until Chapman [18] demonstrated forests could prominently increase TKE under some specific conditions’ and in L364 ‘the KE of a large drop was much greater than the sum of the energies of a number of smaller drops [38], we took the height at the middle of crown (5.1 m) as the average falling height of throughfall raindrops’ respectively.
17.Brandt, C. J. Simulation of the size distribution and erosivity of raindrops and throughfall drops. Earth Surface Processes and Landforms. 1990, 15, 687-698.
18. Chapman, G. Size of raindrops and their striking force at the soil surface in a red pine plantation. Transactions, American Geophysical Union. 1948, 29, 664-670.
38. Brandt, J. The transformation of rainfall energy by a tropical rain forest canopy in relation to soil erosion. Journal of Biogeography, 1988, 15, 41–48.
Point 3: Reviewing is insufficient. Read and cite recent papers on throughfall DSDs, kinetic energy and plot study to measure soil erosion in forest, for example:
- Miyata, S., Kosugi, K., Gomi, T., & Mizuyama, T. (2009). Effects of forest floor coverage on overland flow and soil erosion on hillslopes in Japanese cypress plantation forests. Water Resources Research, 45(6). https://doi.org/10.1029/2008WR007270
- Goebes, P., Schmidt, K., Härdtle, W., Seitz, S., Stumpf, F., Oheimb, G. von, & Scholten, T. (2016). Rule-based analysis of throughfall kinetic energy to evaluate biotic and abiotic factor thresholds to mitigate erosive power. Progress in Physical Geography, 40(3), 431–449. https://doi.org/10.1177/0309133315624642
- Shinohara, Y., Ichinose, K., Morimoto, M., Kubota, T., & Nanko, K. (2018). Factors influencing the erosivity indices of raindrops in Japanese cypress plantations. CATENA, 171, 54–61. https://doi.org/10.1016/j.catena.2018.06.030
- Song, Z., Seitz, S., Zhu, P., Goebes, P., Shi, X., Xu, S., … Scholten, T. (2018). Spatial distribution of LAI and its relationship with throughfall kinetic energy of common tree species in a Chinese subtropical forest plantation. Forest Ecology and Management, 425(March), 189–195. https://doi.org/10.1016/j.foreco.2018.05.046
- Levia, D. F., Nanko, K., Amasaki, H., Giambelluca, T. W., Hotta, N., Iida, S., … Yamada, K. (2019). Throughfall partitioning by trees. Hydrological Processes. https://doi.org/10.1002/hyp.13432
Response 3: I accepted the comment. I have corrected it in L60 ‘However, TKE was influenced by many factors including drop size distribution (DSD) [21-23], canopy architecture (e.g., canopy thickness, height) [24,25], tree species [14,22,25], even meteorological factors [14,22] and so on’ and L421 ‘The ground cover reduced soil detachment by mitigating raindrop impact, and it could be more effective than tree canopy on reducing soil erosion in forest landscapes [15,48]’ respectively.
15.Zhou, G.Y.; Wei, X.H.; Yan, J.H. Impacts of eucalyptus (Eucalyptus exserta) plantation on sediment yield in Guangdong Province, southern China—a kinetic energy approach. Catena. 2002, 49, 231-251.
21. Calder, I.R.; Hall, R.L.; Prasannab, K.T. Hydrological impact of Eucalyptus plantation in India. Journal of Hydrology. 1993, 150, 635-648.
22. Levia, D. F.; Hudson, S. A.; Llorens, P.; Nanko, K. Throughfall drop size distributions: a review and prospectus for future research. Wiley Interdisciplinary Reviews: Water. 2017, 4, e1225.
23. Zabret, K.; Rakovec, Jože; Mikoš, Matjaž; Šraj, Mojca. Influence of raindrop size distribution on throughfall dynamics under Pine and Birch trees at the rainfall event level. Atmosphere. 2017, 8, 240.
24. Nanko, K.; Giambelluca, T. W.; Sutherland, R. A.; Mudd, R.G.; Nullet, M.A.; Ziegler, A.D. Erosion Potential under Miconia calvescens Stands on the Island of Hawai‘i. Land Degradation & Development. 2015, 26, 218-226.
25. Goebes, P.; Schmidt, K.; Ha¨rdtle, W.; Seitz, S.; Stumpf, F.; von Oheimb, F.; Scholten, T. Rule-based analysis of throughfall kinetic energy to evaluate biotic and abiotic factor thresholds to mitigate erosive power. Progress in Physical Geography, 2016, 40, 431-449.
48. Miyata, S.; Kosugi, K.; Gomi, T.; Mizuyama, T. Effects of forest floor coverage on overland flow and soil erosion on hillslopes in Japanese cypress plantation forests. Water Resources Research, 2009, 45, 192-200.
Point 4: L62. There are some studies to estimate throughfall KE under low canopy height. Add references here.
Response 4: I have made some changed in the Introduction, and the sentence ‘There are some studies to estimate throughfall KE under low canopy height’ were deleted.
Point 5: L88-95. Show tree height, canopy bottom height, stand density (or the number of trees trees in the plot), and canopy openness. They are useful for readers.
Response 5: The added information of tree height, canopy bottom height and stand density were shown in L118-119 ‘the average tree height was 7.4 m, the average bottom height was 3.1 m’ and marked in red. There was no information of the canopy openness but the canopy closure which was the ratio of projected canopy area to the plot area. 1 minus canopy closure is canopy openness.
Point 6: Section 2.2.1. Clarify your measurement methodology. Sampling area of the rain gauge, sampling area of the buckets for throughfall measurement, and the number of trees to measure stemflow. Also, did you measure sediment yield without stemflow input in the plot? If so, you should describe it.
Response 6: I have added the information mentioned above. Sampling area of the rain gauge was 0.031 m2, shown in L135. Sampling area of the buckets for throughfall measurement was shown in L138 ‘The top area of the bucket was 0.035 m2 with length 0.25 m and width 0.14 m’. The number of trees to measure stemflow was shown in L141-142 ‘Three masson pines with similar DBH (Table 2) were selected in masson pine plots for stem flow sampling’. Considering that stem flow accounts for a very small proportion of precipitation, and it has little effect on soil erosion, we did not measure sediment yield without stemflow input. The proportion of stem flow to precipitation we obtained in the paper was only 1.6%.
Point 7: Section 2.2.1. Clarify your calculation methodology. Did you use mean of three throughfall buckets? If so, how much was the variability? How did you convert stemflow volume [L] into [mm]?
Response 7: The variable coefficient of throughfall was 1.6-15.9%, shown in L140. The variable coefficient of stem flow was 2.0-20.0%, shown in L146. The added method of volume conversion was shown in L144-146 ‘The volume of collected rainwater measured by graduated cylinder with 1 ml accuracy divided by bottom area of the bucket is equal to stem flow’.
Point 8: Section 2.2.2. You selected totally 48 points (16 points x 3 trees) for the filter paper method. How many stain papers did you use in this study? The detail is not shown.
Response 8: The number of stain paper we use in this study was 662 for bare plot, 2528 for forest plot, shown in L176-177. Because of the large number of the stain paper for the forest plot, some of them were not dried in time and some raindrops were overlapped, the actual number of the stain paper was different from the theoretical value.
Point 9: The total rainfall amount was 176.2 mm for your measurement, however is was SMALLER than annual rainfall, 1700 mm. Did the observed rainfall events (Table 2) have average characteristics? Do they have representativeness for this area? I cannot judge from the paper because characteristics of rainfall events in this area, such as ranges of average rainfall intensity, rainfall amount and rainfall duration, are not shown. For example, if you collected rainfall events with relatively smaller intensity or amount, your data will overestimate interception loss.
Response 9: I have explained the representativeness of the rainfall events we measured shown in L128-131 ‘Data from the Changting hydrographic bureau 10-year rainfall events showed that events with precipitation < 20 and 35 mm comprised 82.8% and 91.8% percent of the total rainfall events. Events with rainfall intensity < 20 and 45 mm comprised 92.9% and 97.4% percent of the total rainfall events. Therefore, these events we measured were typical’.
Point 10: L258-262. This analysis is interesting. I recommend to add figure or table to show the importance of large drops on sediment yield.
Response 10: I have added a table (Table 5) which showed the correlation between KE of different classes of raindrops and soil loss.
Table 5. The correlation between KE of different classes of raindrops and soil loss
Total KE/J m-2 | Soil loss in bare plot/g | Soil loss in forest plot/g |
Small raindrops (<1 mm) | y = 5950.5 exp (-0.474x), R2=0.72 | y = 3672.3 exp (-0.967x), R2=0.56 |
Medium raindrops (1-3 mm) | y = 31.0 exp (0.022x), R2=0.76 | y = 38.7 exp (0.032x), R2=0.17 |
Large raindrops (>3 mm) | y = 671.2 exp (0.009x), R2=0.94 | y = 110.5 exp (0.016x), R2=0.80 |
Point 11: Section 4.1. Your calculation is based that all of throughfall was falling from 5.1 m high. It underestimates falling velocity of throughfall because (1) free throughfall is falling with terminal velocity, (2) release throughfall is not always fallen from crown bottom but higher part of crowns (see references). The discussion is insufficient.
- Brandt, C. J. (1990). Simulation of the size distribution and erosivity of raindrops and throughfall drops. Earth Surface Processes and Landforms, 15, 687–698. Retrieved from http://onlinelibrary.wiley.com/doi/10.1002/esp.3290150803/abstract
- Nanko, K., Onda, Y., Ito, A., & Moriwaki, H. (2008). Effect of canopy thickness and canopy saturation on the amount and kinetic energy of throughfall: An experimental approach. Geophysical Research Letters, 35, L05401. https://doi.org/10.1029/2007GL033010
- Wakiyama, Y., Onda, Y., Nanko, K., Mizugaki, S., Kim, Y., Kitahara, H., & Ono, H. (2010). Estimation of temporal variation in splash detachment in two Japanese cypress plantations of contrasting age. Earth Surface Processes and Landforms, 35(9), 993–1005. https://doi.org/10.1002/esp
- Nanko, K., Onda, Y., Ito, A., & Moriwaki, H. (2011). Spatial variability of throughfall under a single tree: Experimental study of rainfall amount, raindrops, and kinetic energy. Agricultural and Forest Meteorology, 151, 1173–1182. https://doi.org/10.1016/j.agrformet.2011.04.006
- Song, Z., Seitz, S., Zhu, P., Goebes, P., Shi, X., Xu, S., … Scholten, T. (2018). Spatial distribution of LAI and its relationship with throughfall kinetic energy of common tree species in a Chinese subtropical forest plantation. Forest Ecology and Management, 425(March), 189–195. https://doi.org/10.1016/j.foreco.2018.05.046
Response 11: I have read the references you listed above carefully, and I have corrected in L360‘In generally, broad-leaved trees have larger D50 than coniferous trees because rainwater could not coalesce readily on the fine leaves [27, 37]’ and L365‘the KE of a large drop was much greater than the sum of the energies of a number of smaller drops [38]’ respectively.
27.Nanko, K.; Onda, Y.; Ito, A.; Moriwaki, H. Spatial variability of throughfall under a single tree: Experimental study of rainfall amount, raindrops, and kinetic energy. Agricultural and Forest Meteorology. 2011, 151, 1173-1182.
37. Song, Z.; Seitz, S.; Zhu, P.; Goebes, P.; Shi, X.Z.; Xu, S.X.; Wang, M.Y.; Schmidt, K.; Scholten, T. Spatial distribution of LAI and its relationship with throughfall kinetic energy of common tree species in a Chinese subtropical forest plantation. Forest Ecology and Management, 2018, 425, 189-195.
38. Brandt, J. The transformation of rainfall energy by a tropical rain forest canopy in relation to soil erosion. Journal of Biogeography, 1988, 15, 41-48.
Point 12: L108. Bottom area is not correct. Top area is correct.
Response 12: I have corrected it as shown in L139. But the data was not changed owing to the shape of the bucket used to collect throughfall was rectangular which bottom area was the same as top area.
Point 13: L118. What sizes did you use? Show them concretely.
Response 13: Medical needles with 0.45mm, 0.6mm, 0.7mm, 1.2mm, 1.6mm with different sizes were used to simulate different diameters of raindrops. I have added the sizes of medical needles used to simulate different diameters of raindrops in L156-157.
Point 14: L122. What is “dw”? No explanation.
Response 14: I have made an explanation of the word ‘dw’ in L162: dw is the diameter of a water drop.
Point 15: L132. What was measured at “two sampling points”?
Response 15: All of the 16 points were used to raindrops sampling. Maybe the meaning was not clear in the paper. I have corrected in L170-173: We selected 16 sampling points under each masson pine which were distributed in 8 directions (Figure 2). There were two points in each direction evenly established at the canopy projection onto the ground.
Point 16: Table 2. Is “Height at the middle of crown” is the height between soil surface and crown bottom?
Response 16: The height at the middle of crown was explained in the notes of Table 2. It was the distance from canopy centre to the soil surface.
Point 17: Figure 3. “a, b, c” are not shown in the figure. Caption of x-axis is “event average rainfall intensity”. Add error bar on each plot.
Response 17: I have deleted the “a, b, c” shown in the figure, changed the caption of the x-axis and added error bar on each plot.
Point 18: Figure 4. Use same range for x-axis, such as from 0 to 7 mm.
Response 18: I have changed the x-axis to the same range in Figure 4.
(a)
(b)
Point 19: Figure 6. Power function is not appropriate unit kinetic energy reach plateau (van Dijk et al. 2002). Exponential is better.
Response 19: I have changed it to exponential function. A power law relationship predicts kinetic energy contents at lower rainfall intensity well but overestimates kinetic energy at higher intensities. An exponential curve fits the data particularly well at higher intensities (above 50 mm h-1). In hence, although a power law maybe fits better based on the date we measured, an exponential curve was more practical because higher rainfall intensities are much more important in determining overall storm energy than lower intensities. Thank you for your comment.
Figure 6. Scatter plot of KE versus rainfall intensity with exponential-law

Reviewer 2 Report
p { margin-bottom: 0.25cm; line-height: 115%; background: transparent none repeat scroll 0% 0%; }
General:
By dealing with the modification of erosive rain drop energy
by tree canopies the paper covers an important topic, also because
it also concerns all agroforestry aopproaches. The paper is well
structured and concise based on a suitable experimental design and
well chose methods.
However, several improvements have to be done in clarity of formulations and the contents. Furthermore I would expect a more intensive coverage of the existing scientific literature on modification of rain drop diameter by canopy passage and the effects on erosivity. Soil detachment results should be discussed considering potentially different K-factors (erodibility) of the the top soil of free land and forest. Discussion should be less attached to single numbers of the results, but more to the existing literature and what is new from the results.
Introduction:
Please amend this chapter with more theoretical aspects of modification of rain-drop energy by canopy passage and add more citations (I found first authors Armstrong, Vis, Hall, Nanko, Mosley and more)
L.35 I recommend to specify the “red soils” with an adequate soil classification term according to World Reference Base for Soil Ressources.
L.42: The sentence remains unclear: Do you mean “plant canopy as the first layer that rainfall reaches ..”?
L.46: Abbreviations have to be explained when used first.
L.55: Botanic names have to be given for plant species.
L.62-70: The message of this paragraph remains somewhat obscure. Is it an hypothesis concerning the falling height? Why the a dense canopy of a stand reduces volume yield? What means here “irreplaceable”? Please condense and give the paragraph a focus.
Material and methods:
L.84: see L.35 comment.
L.86: Give data for soil erodibility, mainly texture and soil organic matter, best would be an estimation of USLE k-factor
L88: Do both sites reveal similar soil erodibility?
L.100- L103: I do not understand: Both sites revealed important differences in rainfall characteristics?
Formula 1: Probably it should be the 3rd root. Furthermore, this calculation is trivial. MOre interesting would be the control estimation based on the stain distributionand the regression as a graphic.
Formulas: Improve the typesetting of the formulas!
Results:
Percentage of throughfall and of stemflow are not well fitted from an statistical point of view , because they more reveal a completely erratic results below 7mm/h and then more or less a plateau instead of a continuous increase, this should be mentioned critically in the description.
L.256: It could be helpful to use a the EI30 parameter as an integrating measure for soil erosivity instead of correlating diameter classes with soil detachment. My suggestion is to integrate erosivity in one parameter and compare open land and forest a) for EI30 and b) for detachment (EI30 is known as mostly linear to soil detachment).
Discussion:
In the present form the discussion is a more a continuation of the results part. It should be focused on answering the questions asked in the introduction and on the hypotheses. REmove so detailed analysis of numbers. It could be helpful to include schematic graph that explains the newly found relationships, especially when they deviate from the literature.
L.274-297: Until here I thought the focus of the paper is the modification of rain-drop energy by the canopy. What you discuss here is neither new not really helpful, the theory of raindrop acceleration is well known. Please focus your discussion on specific effects of your pine (please give the botanical name) and compare it to data from the literature (to my knowledge most stem from tropical rain forests with higher canopies).
Conclusions:
In the conclusions no numbers should be repeated, instead you should focus on a) what do your analyses mean to the management concepts and b) what is still missing. Conclusions are no second abstract.
Abstract: is o.k.
Author Response
Thank you very much for your valuable suggestions on this paper, and positive comments on the experimental design and methods. We have added scientific literatures on modification of rain drop diameter by canopy passage and the effects on erosivity and made the revisions on the Discussion part. The following are detailed responses to the comments.
Point 1: Introduction: Please amend this chapter with more theoretical aspects of modification of rain-drop energy by canopy passage and add more citations (I found first authors Armstrong, Vis, Hall, Nanko, Mosley and more)
Response 1: I have added some references in L53-61 in the manuscript with ‘Track Changes’ mode. The following line numbers were all in the manuscript with ‘Track Changes’ mode.
In forests, the canopy changed rainfall kinetic energy (KE) by intercepting precipitation and thus modifying raindrop size and velocity [14]. Throughfall kinetic energy (TKE) is a usually used indicator to express the potential of rainfall erosivity and predict soil erosion rates [15,16]. Brandt [17] firstly developed a model to calculate TKE and it was widely assumed that a forest canopy mitigated rainfall KE until Chapman [18] demonstrated forests could prominently increase TKE under some specific conditions.
14. Nanko, K.; Hotta, N.; Suzuki, M. Evaluating the influence of canopy species and meteorological factors on throughfall drop size distribution. Journal of Hydrology. 2006, 329, 422-431.
15. Zhou, G.Y.; Wei, X.H.; Yan, J.H. Impacts of eucalyptus (Eucalyptus exserta) plantation on sediment yield in Guangdong Province, southern China—a kinetic energy approach. Catena. 2002, 49, 231-251.
16. Liu, J.Q.; Liu, W.J.; Li, W.X.; Jiang, X.J.; Wu, J.E. Effects of rainfall on the spatial distribution of the throughfall kinetic energy on a small scale in a rubber plantation. Hydrological Sciences Journal. 2018, 63, 1-13.
17. Brandt, C. J. Simulation of the size distribution and erosivity of raindrops and throughfall drops. Earth Surface Processes and Landforms. 1990, 15, 687-698.
18. Chapman, G. Size of raindrops and their striking force at the soil surface in a red pine plantation. Transactions, American Geophysical Union. 1948, 29, 664-670.
Point 2: L.35 I recommend to specify the “red soils” with an adequate soil classification term according to World Reference Base for Soil Resources.
Response 2: I have added the soil type based on WRB shown in L35-36: Red soils, classified as Plinthosols in the World Reference Base for Soil Resources and characterized by acidic, nutrient deficient, poor in organic matter and high erodibility are widely studied in the world.
Point 3: L.42: The sentence remains unclear: Do you mean “plant canopy as the first layer that rainfall reaches ”?
Response 3: I have made some changes in Introduction, the sentence ‘plant canopy as the first layer that rainfall reaches’ was deleted.
Point 4: L.46: Abbreviations have to be explained when used first.
Response 4: All the abbreviations have been explained when used first. See L.52, L.53 and L.303.
Point 5: L.55: Botanic names have to be given for plant species.
Response 5: we have added the botanic names where they first appearance. See L.17,L.32, L.46.
Point 6: L.62-70: The message of this paragraph remains somewhat obscure. Is it an hypothesis concerning the falling height? Why the dense canopy of a stand reduces volume yield? What means here “irreplaceable”? Please condense and give the paragraph a focus.
Response 6: I have made some changes in Introduction, the sentence ‘many previous studies were undertaken either where the height of forest canopy was enough to accelerate raindrops to their terminal velocity …’ was deleted. The ‘irreplaceable’ has been explained in L.84: As a zonal and good pioneer tree for forestation, masson pine is an irreplaceable coniferous afforestation species in the red soil region of southern China.
Point 7: L.84: see L.35 comment.
Response 7: The “red soil” has been explained in L35-36, so I did not make a further explanation in here.
Point 8: L.86: Give data for soil erodibility, mainly texture and soil organic matter, best would be an estimation of USLE k-factor
Response 8: I have added the information of soil organic matter and soil texture shown in L108-109 ‘The content of soil organic matter ranged from 0.95 to 8.51 g kg-1. The soil texture was mainly sand which content was more than 50%’. Because of we did not test the specific content of sand, silt and clay, K factor could not be calculated in the paper. However, the landform, the low content of soil organic matter and the high content of sang also indicated the area was extremely vulnerable to erosion.
Point 9: L88: Do both sites reveal similar soil erodibility?
Response 9: The rainfall characteristics of the two sites were the same due to their short distance, no mountains between them and no microclimate.
Point 10: L.100-L103: I do not understand: Both sites revealed important differences in rainfall characteristics?
Response 10: The rainfall characteristics of both sites were the same. However, it changed after the canopy, because of the canopy interception. The sentence ‘The driving force of soil erosion was mainly from a short period of rainfall and high intensity on a natural bare slope, however, it is not clear in the forest’ was deleted.
Point 11: Formula 1: Probably it should be the 3rd root. Furthermore, this calculation is trivial. More interesting would be the control estimation based on the stain distribution and the regression as a graphic. Formulas: Improve the typesetting of the formulas!
Response 11: I have corrected Formula 1.
Point 12: Results: Percentage of throughfall and of stemflow are not well fitted from a statistical point of view, because they more reveal a completely erratic results below 7mm/h and then more or less a plateau instead of a continuous increase, this should be mentioned critically in the description.
Response 12: I have corrected them shown in L.219-225: Amounts of throughfall were greater with higher rainfall intensity, additionally the growth rates were almost stable which can be fitted by a power function (Figure 3a). Amounts of stem flow and canopy interception first increased with increasing rainfall intensity, meanwhile the growth rates gradually decreased with the increased levels of rainfall intensity and then tended to be constant which can be fitted by a log function (Figure 3b-c). Percentage of throughfall and stemflow were erratic below 7mm h-1 and then more or less a plateau (Figure 3a-b).
Point 13: L.256: It could be helpful to use a the EI30 parameter as an integrating measure for soil erosivity instead of correlating diameter classes with soil detachment. My suggestion is to integrate erosivity in one parameter and compare open land and forest a) for EI30 and b) for detachment (EI30 is known as mostly linear to soil detachment).
Response 13: Thank you for your good suggestion. The ‘E’ in the EI30 parameter is the total kinetic energy. It is not appropriate to analyse the correlation between kinetic energy of different classes of raindrops and soil loss. Therefore, your good suggestion is what I am going to do in the next study, and I also explained it in the conclusion ‘In the future research the combination of rainfall erosivity EI30 and soil erosion is quite necessary’.
Point 14: Discussion: In the present form the discussion is a more a continuation of the results part. It should be focused on answering the questions asked in the introduction and on the hypotheses. RE move so detailed analysis of numbers. It could be helpful to include schematic graph that explains the newly found relationships, especially when they deviate from the literature.
L.274-297: Until here I thought the focus of the paper is the modification of rain-drop energy by the canopy. What you discuss here is neither new nor really helpful, the theory of raindrop acceleration is well known. Please focus your discussion on specific effects of your pine (please give the botanical name) and compare it to data from the literature (to my knowledge most stem from tropical rain forests with higher canopies).
Response 14: I have made some changes in the discussion. I made a comparison with other species, made a discussion about the falling height, and also made an explanation to the future research. Maybe many trees from tropical rain forests were with higher canopies, but the masson pine in the red soil region of southern China grew slowly and the height was relatively low (see Table 2) owing to the specific conditions in the area, so the discussion about TKE and height in the area was also important in my opinion. Thank you for your good advice.
Table 2. Measurement index of the selected trees
Tree age/a | Tree height/m | DBH/cm | Height at the middle of crown /m | Crown width/m*m | |
1 | 10 | 7.6 | 7.4 | 5.4 | 3.2m*3.5m |
2 | 10 | 7.5 | 6.9 | 4.8 | 3.1m*3.1m |
3 | 10 | 7.3 | 7.1 | 5.2 | 3.1m*3.2m |
Point 15: Conclusions: In the conclusions no numbers should be repeated, instead you should focus on a) what do your analyses mean to the management concepts and b) what is still missing. Conclusions are no second abstract. Abstract: is ok.
Response 15:The conclusions have been corrected in the manuscript.
The sediment yield was largely depended on rainfall intensity and KE of large raindrops in the area. D50 and the DSD range of throughfall under masson pine was enlarged in comparison with that of open rainfall. TKE was higher than OKE when the rainfall intensity was less than 14 mm h-1. The effect reversed when the rainfall intensity was more than 14 mm h-1. The rule of soil loss under forest was revealed in the paper. In the future research the combination of rainfall erosivity EI30 and soil erosion is quite necessary. We also found that pruning and understory vegetation were important factors to prevent soil erosion during the forest management process.
Reviewer 3 Report
General Remarks
The manuscript in titled “Influence of canopy interception and rainfall kinetic energy on soil erosion under forest” demonstrates the effect of afforestation in southern China on soil erosion under forest and compared to the bare land. The authors have taken up an important topic. The effect of coniferous afforestation on erosion processes and sediment yield have been studied all over the world, however specific knowledge is still needed to understand the mechanism in different conditions and to find the proper solutions in individual situation.
The authors observed that both kinetic energy of raindrops as well as canopy interception were important factors affecting sediment yield. The presented research confirmed the beneficial effects of the afforestation on the reduction of soil erosion.
The authors state as well, that combine canopy structure and soil properties should consider the comprehensive influence mechanism of canopy on sediment yield.
The manuscript is well understandable and easy to follow, however, at places I would think some additions and corrections would be necessary. Please find some suggestion needed.
Specific comments:
Line 42-46: Complete the reference, please.
Line 84: Determination the soil type in accordance with international classification would make the research more transparent for audience. For example the World Reference Base for Soil Resources (WRB) and Soil Survey Division Staff, USDA classification are the global correlation scheme for soil classification and international communication.
Line 92: “2” present as a upper index, please.
Line 103: Complete the information about the period of measurements. Analysis of dry periods preceding the occurrence of rain are an important indicator and would be a valuable complement to the presented research. Were the rains the same on the both sites, in a distance of 1 km (especially to the volume of raindrops)?.
Line 134: Correct the size of letters.
Line 230: The letter “Y” is not needed. Figure 7 should rather be closer to the text, where authors first time informed about the presented results, not in the discussion part.
Lines 251 -253:. The sentence is rather the part of discussion, and should be moved.
Figure 3: It could be presented on one page size.
Figure 5: Please change the sequence on Y axis. The lower value should be presented from the down and the higher value on the top. It could be easier to follow the results and comments.
Author Response
Response to Reviewer 3 Comments
Thank you very much for your affirmation of the study topic and the English language and style of the manuscript and specific suggestions for the deficiencies in this paper. We have revised them one by one. The following is the detailed description of the revision.
Point 1: Line 42-46: Complete the reference, please.
Response 1: I have made some changes in Introduction, the sentence in L42-46 was deleted. But I also have explained the rule of canopy on rainfall kinetic energy in L52-55 ‘In forests, the canopy changed rainfall kinetic energy (KE) by intercepting precipitation and thus modifying raindrop size and velocity [14]. Throughfall kinetic energy (TKE) is a usually used indicator to express the potential of rainfall erosivity and predict soil erosion rates [15,16]’ in the manuscript with ‘Track Changes’ mode. The following line numbers were all in the manuscript with ‘Track Changes’ mode.
14. Nanko, K.; Hotta, N.; Suzuki, M. Evaluating the influence of canopy species and meteorological factors on throughfall drop size distribution. Journal of Hydrology. 2006, 329, 422-431.
15. Zhou, G.Y.; Wei, X.H.; Yan, J.H. Impacts of eucalyptus (Eucalyptus exserta) plantation on sediment yield in Guangdong Province, southern China—a kinetic energy approach. Catena. 2002, 49, 231-251.
16. Liu, J.Q.; Liu, W.J.; Li, W.X.; Jiang, X.J.; Wu, J.E. Effects of rainfall on the spatial distribution of the throughfall kinetic energy on a small scale in a rubber plantation. Hydrological Sciences Journal. 2018, 63, 1-13.
Point 2: Line 84: Determination the soil type in accordance with international classification would make the research more transparent for audience. For example, the World Reference Base for Soil Resources (WRB) and Soil Survey Division Staff, USDA classification are the global correlation scheme for soil classification and international communication.
Response 2: I have added the soil type based on WRB shown in L35-36: Red soils, classified as Plinthosols in the World Reference Base for Soil Resources and characterized by acidic, nutrient deficient, poor in organic matter and high erodibility are widely studied in the world.
Point 3: Line 92: “2” present as a upper index, please.
Response 3: I have corrected it to upper index shown in L116: 100 m2.
Point 4: Line 103: Complete the information about the period of measurements. Analysis of dry periods preceding the occurrence of rain are an important indicator and would be a valuable complement to the presented research. Were the rains the same on the both sites, in a distance of 1 km (especially to the volume of raindrops)?
Response 4: The volumetric moisture content of topsoil preceding the occurrence of rain was an important factor in the paper, I have added the information shown in L131-133: The volumetric moisture content of surface soil preceding the occurrence of rain ranged from 0.126 to 0.257 in the bare plot and from 0.159 to 0.305 in the forest plot. The rainfall characteristics of the two sites were the same due to their short distance, no mountains between them and no microclimate.
Point 5: Line 134: Correct the size of letters.
Response 5: I have corrected the size of letters‘were conducted’ shown in L174.
Point 6: Line 230: The letter “Y” is not needed. Figure 7 should rather be closer to the text, where authors first time informed about the presented results, not in the discussion part.
Response 6: The letter “Y” has been moved. It was Figure 5 showed the information presented and I have changed Figure 7 to Figure 5 shown in L293, so I did not change the position of Figure 7.
Point 7: Lines 251 -253: The sentence is rather the part of discussion, and should be moved.
Response 7: The sentence has been deleted and moved to L348-350: It was also suggested that masson pine canopy had a more obvious mitigation effect on KE under high rainfall intensity by comparison with low rainfall intensity in the study area.
Point 8: Figure 3: It could be presented on one page size.
Response 8: Figure 3 has been presented on one page size.
Point 9: Figure 5: Please change the sequence on Y axis. The lower value should be presented from the down and the higher value on the top. It could be easier to follow the results and comments.
Response 9: The sequence on Y axis have been changed.
Figure 5. Volume ratio of raindrops with different size raindrops under different rainfall intensity. The dashed red lines indicate the volume ratio of large drops, and the dashed black lines represent the volume ratio of small drops.

Round 2
Reviewer 1 Report
The paper has been well revised according to reviewers' comments. It is acceptable for publication.
Reviewer 2 Report
The paper has strongly improved. The abstract gives a clear scientific message. My suggestions have been mostly included in the study. Therefore I recommend the publication of the manuscript, probably with some polishing of English and formatting.